# Study of Physicochemical Quality and Organic Contamination in Algerian Honey

**DOI:** 10.3390/foods13091413

**Published:** 2024-05-04

**Authors:** Sofiane Derrar, Vincenzo Lo Turco, Ambrogina Albergamo, Benedetta Sgrò, Mohamed Amine Ayad, Federica Litrenta, Mohamed Said Saim, Angela Giorgia Potortì, Hebib Aggad, Rossana Rando, Giuseppa Di Bella

**Affiliations:** 1Laboratoire d’Hygiène et Pathologie Animale, Institut des Sciences Vétérinaires, Université de Tiaret, Tiaret 14000, Algeria; sofiane.derrar@univ-tiaret.dz (S.D.); mohamedamine.ayad@univ-tiaret.dz (M.A.A.); docsaim2@gmail.com (M.S.S.); hebib.aggad@univ-tiaret.dz (H.A.); 2Department of Biomedical, Dental, Morphological and Functional Images Sciences (BIOMORF), University of Messina, Viale Annunziata, 98122 Messina, Italy; vloturco@unime.it (V.L.T.); felitrenta@unime.it (F.L.); angela.potorti@unime.it (A.G.P.); rrando@unime.it (R.R.); gdibella@unime.it (G.D.B.); 3Department of Chemical, Biological, Pharmaceutical and Environmental Sciences, University of Messina, Viale F. Stagno d’Alcontres 31, 98166 Messina, Italy

**Keywords:** Algerian honey, beekeeping, physicochemical parameters, organic contamination, pesticides, PAHs, PCBs, plasticisers, persistent organic pollutants

## Abstract

Honey is a natural product extensively consumed in the world for its nutritional and healthy properties. However, residues of pesticides and environmental contaminants can compromise its quality. For this reason, the physicochemical parameters, and the organic contamination of monofloral and multifloral honey from three regions of Algeria (Tiaret, Laghouat, and Tindouf) were monitored to evaluate the quality of the honey and its safety for consumers. In general, the results obtained from the physicochemical analyses were in line with the EU standards. In terms of contamination, pesticides authorised and used in Algerian agriculture (metalaxyl-M and cyromazine), as well as a banned pesticide (carbaryl), were found in almost all the samples. However, only the concentration of cyromazine was higher than the relative EU maximum residue levels. PCB 180, PCB 189, anthracene, fluorene, and phenanthrene were mainly detected. All the honey shows traces of DiBP, DBP, DEHP, and DEHT, but no traces of bisphenols were found. Moreover, according to the dietary exposure assessment, a small amount of Algerian honey can be safely consumed. Overall, the data from this study should motivate the Algerian government to enhance their monitoring activities in beekeeping and to find solutions for implementing more sustainable agricultural practices harmonising with international legislation.

## 1. Introduction

Honey is a natural, complex, and plant-based sugar product produced by honeybees (*Apis mellifera* L.) and extensively consumed in the world not just as a direct food but also as a natural flavouring and sweetener agent [1]. Honey mainly consists of fructose and glucose but also includes fructo-oligosaccharides and numerous enzymes, vitamins, minerals, phenolic acids, and flavonoids that provide it with healthy and therapeutic properties [2,3]. The quality of honey relies both on the botanical source of pollen, nectar, and honeydew and on the production context, which includes the quality of the soil, water, and air and the presence of chemical pollutants as well [4]. Environmental contaminants from agricultural, industrial, and urban areas are ingested by bees when they collect nectar, pollen, and honeydew from flowers and drink water, or the contaminants accumulate in their bodies via contact with contaminated surfaces. As a result, environmental pollutants are transferred within hives and accumulated in the honey, which can be considered a potential indicator for environmental contamination [5]. However, to have the most benefit, honey must be free from any type of contaminant.

Residues of pesticides from environmental sources or agriculture and beekeeping practices, such as such as organochlorine (OC), organophosphorus (OP), and carbamates, have been detected in honey, thereby compromising food safety [6]. The European Community Regulation No. 396/2005, as amended, established the maximum allowable residue levels (MRLs) of pesticides in foods, including honey [7].

The presence of many persistent organic pollutants (POPs) in honey samples is well recognised. POPs are considered a serious problem because they accumulate in the environment and human body due to their ubiquity, lipophilicity, propensity of bioaccumulation, biomagnification, and persistence in the environment [8].

Among POPs, OC pesticides, polycyclic aromatic hydrocarbons (PAHs), and polychlorinated biphenyls (PCBs) are the most notorious pollutants involved in the emergence of several health problems [9].

Due to the persistence and adverse effects on human health using OCPs and PCBs, some of these chemicals were banned or limited in many countries by the Stockholm Convention on Persistent Organic Pollutants, adopted in 2001, entering into force in 2004 [10]. In 2005, the European Union identified 16 PAHs that possess both carcinogenic and genotoxic properties, 12 of which are also identified to be definite carcinogens for humans by the International Agency of Cancer (IARC) [11]. Nevertheless, due to uncontrolled and indiscriminate anthropogenic activities, residues of OCPs from agricultural practices, and traces of PCBs and PAHs are still persistent in the environment, accumulate in plants from polluted soil, and pass into food, honey included [12,13].

In addition to environmental pollution, other sources of honey contamination are represented by beekeeping activities, honey production, and packaging processes. Plastic additives, such as bisphenols (BPs), phthalates (PAEs), and non-phthalate plasticisers (NPPs), have been found in honey [14]. The cause of this contamination type is mainly related to the direct contact between honey and unsuitable plastic during production and storage. In addition, recently, plastic honeycombs have been used to reduce the risk of melting the wax itself, resulting in reduced yield [15]. The European Union (EU) and the United States Environmental Protection Agency (USEPA) have listed six phthalates as priority toxic pollutants [16]. In 2023, the EFSA released the scientific opinion of public health risks associated with the presence of BPA in food, thus regulating its Tolerable Daily Intake (TDI) at 0.2 ng/kg_bw_/day [17].

Many of these organic pollutants (i.e., DDT, atrazine, chlorpyrifos, PCBs, some PAEs, and BPA) are also classified as Endocrine-Disrupting Chemicals (EDCs) because of the capacity to interfere with the synthesis, release, transport, binding, or removal of natural hormones in the human body, resulting in carcinogenic, mutagenic, and teratogenic effects [18,19]

It is essential to monitor residues of contaminants in foodstuffs, such as honey, to prevent human health risks.

Algeria is considered a traditional consumer of honey, but national production does not achieve self-sufficiency because of the lack of national legislation and the rural organization of this ancient practice [20]. The northern region of Algeria, characterised by a Mediterranean climate and great diversity of flora, lends itself to beekeeping, while the high steppe plateau and the large Saharan plateau in the south of Algeria are less suitable for beekeeping [21]. In 2021, the national honey production was estimated to be 5165 t, with a yield of 4 to 8 kg per hive, which is very low considering the potential offered by Algeria and the 150,000 tons of honey per year imported from other countries [22,23]. In the absence of national legislation, there are no criteria to check the safety and the quality of Algerian honey.

Considering this scenario, the aim of the present study was to investigate the physicochemical parameters of honey (i.e., moisture, total soluble solids, pH, electrical conductivity, and acidity) and the presence of organic chemical residues in samples from different areas of Algeria. The aim was to monitor the quality and safety of honey from different geographical areas and to assess dietary exposure to contaminants by exploiting the Algerian guidelines.

## 2. Materials and Methods

### 2.1. Honey Sample Collection

A total of 54 honey samples were collected during 2022 and 2023 by beekeepers located in three different provinces of Algeria (i.e., Tiaret, Laghouat, and Tindouf), as detailed in Table 1. Samples with the same geographical and botanical origins were grouped together.

Laghouat and Tiaret are two of the most famous Algerian regions for their production of honey [23]. Laghouat is an ancient oasis in the southern foothills of the Saharan Atlas, characterized by the presence of flora very similar to that present in Mediterranean regions [24]. Tiaret is situated in the western steppe region of Algeria, and, in the north, there are dense forest areas that contain many different species of plants [25]. Tindouf, located in the natural region of the Sahara Desert, is characterised by the low diversity and abundance of plant species due to the extreme environmental conditions [26]. Figure 1 shows the geographical map of the sampling sites considered for the study.

The honey samples obtained from these areas of Algeria were collected in glass containers of approximately 150 g and stored in a dark place at ambient temperature until analysis.

### 2.2. Chemicals and Reagents

All solvents and reagents were of analytical grade. Ultrapure water HPLC-grade and acetonitrile (purity ≥ 99.9%), and n-hexane were purchased from Merck (Darmstadt, Germany).

A total of 108 pesticides, 18 PCBs, and 13 PAHs standards were provided from Fluka Analytical (Milan, Italy), Dr. Ehrenstorfer (Augsburg, Germany), and Aldrich Chemical (Chicago, IL, USA). The deuterated analogues used as internal standards (ISs) for pesticide analysis (atrazine-d5, carbofuran-d3, cyprodinil-d5, dimethoate-d6, imazalil-d5, malathion-d6, methiocarb-d3, and trifloxystrobin-d6) were all purchased from Toronto Research Chemicals Inc. (North York, Canada), while the deuterated analogues for PCB analysis (acenaphtene-d10, naphtalene-d8, and phenanthrene-d10) were obtained from Cambridge Isotope Laboratories Inc. (Andover, MA, USA). Ten PAE and eight NPP analytical standards (purity ≥ 96%) were obtained from Supelco (Bellefonte, PA, USA). Dibutyl phthalate-d4 (DBP-d4) and bis(2-ethylhexyl)phthalate-d4 (DEHP-d4) in nonane were used as ISs and purchased from Cambridge Isotope Laboratories Inc.

Stock solution of each pesticide, PCB, PAE, and NPP were prepared at a concentration of 1000 mg/L in n-hexane and subsequently stored at 4 °C.

BP (*n* = 9) analytical standards (purity ≥ 99%) were provided by Sigma-Aldrich (Bornem, Belgium), while the ISs ^13^C_12_-BPA and ^13^C_12_-BPS (purity ≥ 99%) were obtained from Cambridge Isotope Laboratories. Stock solutions of BPs were prepared at 100 mg/L in acetonitrile and stored at 4 °C.

The Quick, Easy, Cheap, Effective, Rugged, and Safe (QuEChERS) Q-sep extraction kits (4 g MgSO_4_ + 1 g NaCl and 6 g MgSO4 + 1.5 g of CH_3_COONa), d-SPE (750 mg MgSO_4_ + 250 mg of primary and secondary amines PSA + 125 mg of octadecyl sorbent C18), and clean-up kit (25 mg C18) were purchased from Agilent Technologies Italia S.p.A. (Milan, Italy).

To significantly limit the contamination caused by laboratory materials and solvents, sample preparation time, laboratory equipment, and solvent contact with samples and solvent volumes used were minimised as much as possible. Stainless steel instruments and glassware underwent initial washing with acetone followed by n-hexane, were dried at 400 °C for 4 h, and finally covered with aluminium foil until analysis.

### 2.3. Physicochemical Parameters

The physiochemical parameters of each honey sample were calculated according to official methods [27].

The moisture (expressed as a percentage) and the total soluble solids (TSSs), which is the concentration of soluble sugars (expressed as °Brix), were determined by means of an Abbe refractometer. The values derived from conversion tables correlating water content and °Brix with the refractive index were determined for each sample at a temperature of 20 °C. In cases where the refractive index was measured at a temperature other than 20 °C, adjustments were made to standardize the results to this temperature.

Free, combined, and total acidity were calculated using the titrimetric method. Briefly, 10 g of each honey sample diluted in 75 mL of pure water was titrated with 0.05 M of NaOH until reaching pH 8.5 (indicating free acidity). Subsequently, 10 mL of NaOH was added, and titration was continued with 0.05 M of HCl until reaching pH 8.3 (indicating combined acidity). The total acidity was determined by summing the values obtained for free and combined acidities.

The electrical conductivity and pH of each honey sample were measured using an Oakton PC 2700 Benchtop conductivity/pH meter (Cole-Palmer, Vernon Hills, IL, USA). About 10 g of honey was dissolved in 75 mL of pure water, and the electrical conductivity and pH were recorded. To determine the amount of honey needed for electrical conductivity measurement, the Formula (1) below was used:(1)$$M=20\times \frac{100}{100-A}$$
where *M* is the quantity of honey expressed in grams to be weighed; 20 is the theoretical nominal mass of honey; and *A* is the moisture in %.

### 2.4. Pesticide, PCB, and PAH Residues

The method adopted for the extraction of pesticides, PCBs, and PAHs from honey samples, as previously validated by Massous et al. [28], is explained below. Briefly, 10 g of each honey sample was measured into a centrifuge tube, dissolved with 10 mL of pure water and 10 mL of acetonitrile, and then vortexed for 5 min. Then, a Q-sep QuEChERS extraction kit (4 g MgSO_4_ + 1 g NaCl) and d-SPE (750 mg MgSO_4_ + 250 mg of PSA + 125 mg of C18), described in Section 2.2, were added and, after shaking vigorously for approximately 1 min, each sample was centrifuged for 10 min at 4 °C at 8000 rpm. Finally, 5 mL of the organic phase was reduced to 1 mL in a rotary evaporator at 30 °C and, in the end, to 0.5 mL under a stream of nitrogen. A known amount of bromophos-methyl as IS was added to each sample before the instrumental analysis,

The multiresidue analysis was carried out using a Shimadzu GCMS-TQ8030 triple quadrupole mass spectrometer (Shimadzu, Kyoto, Japan). The GC-MS conditions are reported in Table 2.

The identification of pesticides, PAHs, and PCBs was carried out by comparing their mass spectra and retention times to those of corresponding commercially available standards. The Multiple Reaction Monitoring (MRM) mode was used for the quantification of analytes, exploiting the IS normalisation, according to our previous study [28]. The MRM transitions, as well as the analytical method validation, are reported in Appendix A. The LabSolutions software 4.01 (Shimadzu) was used for data acquisition and quantification. Each honey sample was analysed three times, along with analytical blanks, to ensure the accuracy and reliability of the measurements.

### 2.5. Plasticiser Residues

The extraction method of plasticiser residues from honey samples as reported in Massous et al. [28] was adopted in the present work. Quickly, 5 g of each honey sample was measured in a centrifuge tube; dissolved with 10 mL of acetonitrile; combined with Q-sep QuEChERS (4 g MgSO_4_ + 1 g NaCl), as described in Section 2.2; and centrifuged for 10 min at 8000 rpm. Then, about 2 mL of the organic phase was reduced to 1 mL in a rotary evaporator at 30 °C and, finally, to 0.5 mL under a stream of nitrogen. A known amount of DEHP-d4 and DBP-d4 was added to each sample before the instrumental analysis. The identification of plasticisers was carried out using a Shimadzu GCMS-TQ8030 triple quadrupole mass spectrometer (Shimadzu, Kyoto, Japan). Table 2 reports the operative conditions. PAEs and NPPs were identified by comparing their mass spectra and retention times to those of commercially available standards. Quantitative analysis was carried out in Single-Ion Monitoring (SIM) mode, taking into account the base peak ion among three characteristic mass fragments for each target analyte, reported in Appendix A, and employing internal standard (IS) normalisation, as suggested by Liotta et al. [29]. The method validation is detailed in Appendix A. The LabSolutions software 4.01 (Shimadzu) was used for data acquisition and quantification. Each sample was analysed in triplicate with analytical blanks, to ensure the accuracy and reliability of the measurements. Glass equipment was used to avoid plasticiser contamination.

### 2.6. BP Residues

The Micro-QuEChERS procedure, developed and validated by Potortì et al. [30], was followed to extract and detect bisphenol analogues in the honey samples. Briefly, 1.5 g of honey was mixed with 3 mL of ultrapure water and 100 μL of ^13^C_12_BPA in a glass tube, widely shaken, and left in a dark place for 24 h prior to the extraction procedure. Then, the QuEChERS Q-sep extraction Kit (MgSO_4_ and CH_3_COONa) and clean-up kit (C18) were used with 3 mL of acetonitrile as the extraction solvent. Subsequently, 1 mL of acetonitrile extract was filtered through a PVDF syringe filter (0.22 μm) and injected into a Shimadzu UHLPC-MS/MS system (Shimadzu, Tokyo, Japan). The apparatus consists of a triple quadrupole mass spectrometer (LCMS-8040), a degasser (DGU-20A5R), an autosampler (SIL-30AC), a column oven (CTO-30A), two pumps (LC-30AD), a controller (CBM-20A), and gradient valve regulators. The instrumental operating conditions are indicated in Table 3. The data were obtained using MRM mode and the ion transitions were used to identify the analyte. The quantification was carried out using the IS method. MRM transitions and analytical validation metrics for each target analyte are detailed in Appendix A. The LabSolutions software 5.75 (Shimadzu) was used for data acquisition and quantification. Each honey sample was analysed in triplicate with analytical blanks, to the ensure accuracy and reliability of the measurements. Glass equipment was used to avoid bisphenol contamination.

### 2.7. Statistical Analysis

Experimental data are presented as the means ± standard deviation of three replicate measurements for each sample.

The statistical analysis was performed using the SPSS 13.0 software package for Windows (SPSS Inc., Chicago, IL, USA). One-way analysis of variance (ANOVA) was used for each independent variable to show statistically significant differences. When a significant F was determined (Fcalculated > Fcritical), Tukey’s honestly significant difference (HSD) test was performed for all pairwise comparisons of means. For each variable examined, statistical significance was accepted at the level of *p* < 0.01. To identify the differences in organic contaminants in honey from different regions of Algeria, the dataset was standardised to achieve uniform significance for all variables and the principal component analysis (PCA) was performed, including only the contaminants quantified in at least 60% of the samples analysed.

### 2.8. Assessment of the Dietary Exposure to Contaminants

In order to assess the health risks associated with the organic contaminants present in Algerian honey, the relative Estimated Daily Intakes (EDIs) were first determined. EDIs were calculated by multiplying the mean concentration of contaminants quantified in each sample (expressed in mg/kg or μg/kg) by the daily consumption of honey (in grams) and dividing the resulting value by the body weight of the consumers (in kilograms). Additionally, the chronic non-carcinogenic risk derived from dietary exposure to the chemical was calculated in terms of Hazard Quotient (HQ). HQ is obtained by dividing a specific EDI by the corresponding Acceptable Daily Intake (ADI) or Tolerable Daily Intake (TDI). When HQ < 1, there are no health risks to the exposed population.

## 3. Results and Discussions

### 3.1. Physicochemical Parameters

Table 4 shows the values of moisture; TSS; conductivity; pH; and the free, combined, and total acidity of honey from different regions of Algeria.

Moisture is a crucial parameter in controlling the quality of honey because high moisture content can cause its undesirable fermentation [31]. This parameter depends on environmental conditions (i.e., pedoclimatic conditions and soil characteristics), activities conducted by beekeepers, harvest time, and honey maturity level. The moisture influences honey taste, colour, flavour, viscosity, density, crystallisation, and fermentation during storage [32]. In the honey analysed, moisture values were in the range of 12.32–16.55% (*p* < 0.01), lower than the maximum limit set by the Codex Alimentarius and EU regulation (20%) [33,34]. This indicates that all honey samples investigated in this study have reached a good level of maturity.

The TSS of honey reflects the sugar compounds present in honey and it is inversely correlated to the moisture [35]. In the honey analysed, the maximum value was 84.91 °Brix in *E. globulus* honey from Laghouat while the minimum was 81.28 °Brix in multifloral honey samples from Tiaret (*p* < 0.01). High values of TSS contribute to osmotic stress for selected microorganisms [35].

The electrical conductivity of honey depends mainly on mineral content but also on organic acids in honey and it is influenced by the geographic location and botanical source. This parameter is useful in discerning between nectar and honeydew honey because it is generally higher in honeydew honey [36]. In Algerian honey, the conductivity varied from 241.48 μS/cm to 565.43 μS/cm (*p* < 0.01), respectively, in *E. sativa* honey from Tindouf and *Tamarix* and *E. orientalis* honey from Laghouat. The Codex Alimentarius and EU regulation [33,34] fixed the electrical conductivity lower than 800 µS/cm for nectar honey, so this parameter demonstrates that all the samples analysed were honey obtained from nectar.

An acidic character was observed in all analysed samples. A normal pH value for honey falls within the range of 3.2 to 4.5 [37]. pH values of Algerian honey ranged between 3.61 in *E. sativa* honey from Tindouf and 4.86 in *Z. lotus* honey from Laghouat (*p* < 0.01). Both analysed types of *Z. lotus* honey are characterized by a pH value out of the suggested range.

Honey acidity is correlated to pH, and vice versa. Good honey is characterised by high acidity and low pH, parameters that inhibit microorganism growth [38]. Acidity is determined via the content of organic acids (mainly gluconic acid, derived from the enzymatic reaction of glucose oxidase with glucose in the presence of water) and, therefore, via the enzymatic activity, and it is related to the freshness of the honey [39]. The acids are in a fluctuating equilibrium between their free and combined forms represented by lactones. For this reason, the value of the total acidity of honey is given through the sum of free and combined acidity. The maximum level of free acidity is set at 50 meq/kg from the EU and Codex Alimentarius [33,34]. Most of the samples analysed in this study showed a level of free acidity below the limit fixed, specifically in the range between 20.10 and 46.67 meq/kg (*p* < 0.01) in *Z. lotus* from Laghouat and multifloral honey from Tiaret, respectively. The same samples showed the lowest and the highest values of total acidity, respectively (22.95 meq/kg in Z_L_ and 47.54 meq/kg in M_T_). The maximum level of combined acidity is found in the *E. globulus* honey from Laghouat. Only the *E. orientalis* honey from Tindouf showed a level of free acidity higher than the EU limits (52.96 meq/kg). The two analysed types of *Z. lotus* honey, with the higher pH value, also have lower levels of free and total acidity than all the samples.

For all parameters studied, statistically significant differences were found due to the great variability of the botanical and geographical origins of the analysed honey. Only for TTS did Tukey’s HSD test show that there is no significant difference between the means of any pair despite a *p*-value lower than 0.01.

In general, moisture, TTS, conductivity and pH of honey samples evaluated in this study were in line with the results reported in recent studies concerning Algerian honey from different regions [20,23,40]. However, the values of acidity were higher than the data reported in the literature [41,42]. These data show that Algerian honey displays medium quality parameters.

### 3.2. Pesticides, PCBs, and PAHs

Pesticides, PCBs, and PHAs residues revealed in the several honeys from Algeria are shown in Table 5 and Figure 2.

Pesticide residues were detected in all honey samples. Among the pesticides investigated, 16 pesticides were detected: exactly 4 herbicides, 3 carbamates, 2 fungicides, 2 insect growth regulators, 2 OPPs, 2 pyrethroid insecticides, and 1 OCP. *E. orientalis* honey from Laghouat (EO_L_) shows the highest number of quantifiable pesticides (*n* = 9) while the lowest number of pesticides (*n* = 2) was detected in *E. sativa* honey from Tiaret (ES_T_). Cyromazine and metalaxyl-M were detected in all samples, followed by carbaryl, detected in 13 samples. Cyromazine was the pesticide detected in the highest concentration in all samples (in the range from 163.58 µg/kg in *B. mauritanicum* honey from Tiaret to 6.48 µg/kg in *E. orientalis* honey from Tindouf) except for *Echinops* honey from Tindouf, which has the highest concentration of carbaryl (0.30 vs. 15.81 µg/kg). Omethoate and propazine showed concentrations above the LOQ in 50% of the samples analysed. The highest concentration of omethoate was detected in *P. harmala* honey from Tindouf (27.54 µg/kg), while the highest concentration of propazine was found in *B. mauritanicum* honey from Tiaret (1.93 µg/kg). Alachlor was detected in two samples from Tiaret and four samples from Laghouat at concentrations lower than 1 ppb. Bendiocarb, propyzamide, pyriproxyfen, and methidathion were found in only one sample at different concentrations from 0.12 to 3.82 µg/kg. Most of the pesticides found in Algerian honey are not authorised for use in the European Union, except for metalaxyl-M, propyzamide, and pyriproxyfen, according to Regulation (EU) No. 540/2011 [43].

In accordance with the European Community Regulation No. 396/2005 and subsequent revisions [7], the concentration of pesticides was below the maximum residual limits (MRLs), except for cyromazine, which greatly exceeds its MRL of 50 µg/kg in 43% of the investigated samples (M_T_, ZL_T_, BM_T_, TE_L_, EO_L_, and M_L_).

Among the 13 PAHs analysed, 6 compounds were detected. Anthracene, fluorene, and phenanthrene were found in most of the samples. The highest concentration of these compounds was found in *E. globulus* honey from Laghouat (1.55, 5.73 and 2.33 µg/kg of anthracene, fluorene and phenanthrene, respectively). Similar to pesticides, *E. orientalis* honey from Laghouat exhibited the highest level of contamination among the products, with *n* = 9 PAHs detected at a level > LOQ. *E. sativa* honey from Tiaret is the only sample with the concentration of all PAHs under analysis lower than the corresponding LOQ.

Among the 18 PCBs analysed, 6 compounds were detected. PCB 180 and PCB 189 were found in most of the samples at very low concentrations (0.12–0.43 µg/kg, *p* < 0.01). Similar to PAHs, *E. globulus* honey from Laghouat exhibited the highest level of contamination among the products, with n = 4 PCBs detected at a level > LOQ. Among these, PCB 153 (4.67 μg/kg) and PCB 138 (1.59 μg/kg) were the most abundant. In two types of honey from Tindouf (E_D_ and PH_D_), there were no PCB residues. The Regulation (EU) No 915/2023 established the limit for benzo[a]pyrene; the sum of benzo[a]pyrene, benz(a)anthracene, benzo(b)fluoranthene, and chrysene; and the sum of dioxins and dioxin-like PCBs in various foods, excluding honey [44]. Therefore, it is not possible to make toxicological considerations and evaluate the safety of honey samples in relation to these substances.

In the literature, there is no study concerning the organic contamination of Algerian honey. Moreover, the organic contamination of Algerian foods different from honey is little investigated. This was the first study to assess organic contamination in honey samples from Algeria.

The pesticide profile observed in honey samples from Algeria reflects the diversity of agricultural practices. Cyromazine is an insect growth regulator still used in Algeria and, despite no literature data on cyromazine in food being available, the high concentration found in the honey from this study may imply its large use in agriculture. The presence of this insecticide in the environment could be unhealthy for bees, resulting in poor honey production. Also, metalaxyl-M, a fungicide found in all samples, is the active ingredient used in several plant protection products on the Algerian market [45]. However, although carbaryl is one of the 23 substances banned from the Algerian market, residues of this pesticide were found in almost all samples. Referring to the literature data, traces of metalaxyl were found in apples, grapes, nectarines, plums, pears, peaches, and tomatoes from Algeria [46,47]. The widespread adoption and extensive use of pesticides in Algeria are considered essential for controlling pests, diseases, and weeds; minimising or preventing yield losses; and upholding a high level of productivity [48]. The extended persistence of pesticides on plants and soil can indeed lead to issues across the entire food chain. In fact, bees may inadvertently carry these contaminants from plant pollen and nectar back to the hive. As a result, these substances have the potential to be assimilated into various hive products [49].

Concerning PCB and PAH contamination, there are no studies related to Algerian foods. The available studies concern only the contamination of soil and water. Among these, Halfadji et al. indicated that in northwest Algeria, the main sources of PAHs derive from pyrogenic activities and petrogenic contributions, such as coal and wood combustion, fossil fuel and waste incineration, and industrial processes. Additionally, the main origins of PCBs are attributed to commercial PCB mixtures used for industrial applications, including oil-filled insulators and dielectric fluids in transformers and capacitors [50]. In fact, the same study found the presence of PCBs and PAHs in agricultural areas because of their proximity to industrial sites and urban areas.

Recent studies on the monitoring of pesticides, PCBs, and PAHs in honey produced in the Mediterranean area have generally shown differences in the type of contamination with respect to honey from Algeria. No traces of cyromazine were identified in honey from European countries in contrast to honey from Morocco [28]. On the contrary, the presence of OCPs and their toxic metabolites was detected in honey from industrialised areas and intensive apple orchards from Italy [51]. The presence of different pesticides in honey, therefore, depends mainly on the agricultural practices used in different countries, hence the importance of honey as an indicator of environmental pollution. PCB and PAH residues were found in honey from Italy and Turkey, confirming that these contaminants are ubiquitous [12,52,53].

### 3.3. Plasticizers and BPs

Seven PAEs (i.e., DEP, DPrp, DBP, DiBP, BBP, DPhP, and DEHP) and two NPPs (i.e., DEA and DEHT) were detected in honey samples as shown in Table 6 and Figure 3. 

DiBP, DBP, DEHP, and DEHT were determined at a concentration >LOQ in all the samples. The honey samples from Tindouf were characterised by a higher concentration of DiBP (in the range of 0.175–0.266 mg/kg) than the honey from the other two regions (in the range of 0.039–0.070 mg/kg), with statistically significative differences (*p* < 0.01). DEP was the plasticiser found at the highest concentrations in the *Z. lotus* honey from Laghouat (1.656 mg/kg) but the concentrations of this plasticiser were not statistically different (*p* = 0.02). The *Z. lotus* honey from Laghouat was also the sample with the highest number of quantifiable plasticisers (*n* = 7).

To the best of the authors’ knowledge, there is no literature concerning plasticisers in Algerian honey. Since plasticisers can leach from plastic equipment used in honey production processes (such as honey extractors and uncorkers), there is a possibility of honey contamination during the production steps. However, contamination during honey storage can be excluded because the honey was stored in glass jars. Nevertheless, it can be considered that plasticisers are ubiquitous in the environment, so the contamination of the nectar cannot be excluded [54]. In this regard, DEHP is the most frequently identified plasticiser in honey samples [15].

Regarding bisphenols, the concentration of BPA and all its analogues was below the LOQ in all the samples analysed. The only study in the literature on bisphenols in honey from Algeria and Tunisia showed the presence of BPA, BPAP, BPF, BPS, and BPZ residues in Algerian honey at very low concentrations [30].

### 3.4. Principal Components Analysis

In this study, a PCA was exploited to differentiate honey samples by geographical origin based on contamination data. The variables considered in the PCA included only carbaryl, metalaxyl-M, cyromazine, fluorene, PCB 180, PCB 189, DBP, DiBP, DEHP, and DEHT, as these contaminants were found in at least 60% of the samples. All other contaminants were excluded from the PCA, since the inclusion of those contaminants present in <60% of samples would not have allowed the PCA.

Excluding contaminants present in only a few samples, the KMO (Kaiser–Meyer–Olkin) value (0.581) indicates that the data structure is suitable for PCA and the Bartlett’s test of sphericityindicates that the correlations between the variables are not all equal to zero.

Based on the Kaiser Criterion, three principal components (PCs) with eigenvalues greater than 1.0 were extracted. The three components showed a variance of 36.482%, 23.539%, and 13.112%, respectively, for a total of 73.132%.

Figure 4 displays the bidimensional score and loading plots of the first two principal components (PC1 and PC2), which in total explain 60.02% of the variability within the system. To achieve greater system variability, Figure 5 also shows the plots obtained by considering PC1 and PC3.

By overlaying the loading and score plots of Figure 4, variables such as metalaxyl-M, cyromazine, fluorene, PCB 180, and PCB 189 appeared to have a greater influence on the contamination of honey from Tiaret and Laghouat, which indeed exhibited the highest concentrations of these organic contaminants. Moreover, it is clear that honey from Tindouf differed from the others mainly because of the high level of DiBP.

Regarding Figure 5, the loading and score plots show that carbaryl and plasticisers weighed more on Tindouf honey samples, as also demonstrated through the higher carbaryl concentrations in these samples.

Consequently, as we can see from the score plots in Figure 4 and Figure 5, the honey samples from Tindouf tend to cluster separately from the honey from the other two areas, namely Laghouat and Tiaret. However, a separation/differentiation between honeys from these two areas was not possible.

### 3.5. Dietary Exposure to Contaminants

In order to assess the quality of Algerian honey and the potential health risks to consumers, the estimated daily intake (EDI) and non-carcinogenic risk (HQ) of pesticides and plasticisers were calculated, as shown in Table 7. Based on the obtained results, EDIs were calculated by considering the amount of honey consumed daily by a typical-sized adult (70 kg) from Algeria (0.33 g/day) and Europe (1.59 g/day), according to FAO [22]. For the health risk assessment, the HQ for each contaminant detected was less than 1, indicating that the honey is safe for the consumers when ingested at the Algerian and European dietary levels. In fact, the calculated EDIs were well below the ADI for pesticides [55,56,57,58,59,60,61] and the TDI for plasticisers [62], set by international regulatory bodies. This indicates that no adverse health effects result from the consumption of Algerian honey.

## 4. Conclusions

The presence of pesticides, PCBs, PAHs, and plasticisers in food is a global problem. The impact of these substances on bees and their products is significant. Firstly, pesticide residues have a detrimental effect on bees, causing a decline in their population and reducing their ecological services. Secondly, through the contamination of the food chain, negative effects such as endocrine, carcinogenic, reproductive, and neurological effects can affect human life. Increased effort in monitoring and greater public intervention are, therefore, needed to reduce the use of pesticides, PCBs, PAHs, and plasticisers and, hence, minimise the exposure of the whole planet to these substances, including human beings.

In this regard, the characterization of Algerian honey, which includes not only physicochemical parameters but also organic contaminants, was performed for the first time, adding to the limited existing literature on Algerian honey. In terms of physicochemical parameters, the honey analysed complied with the parameters set by the European Union EU to guarantee the authenticity of these bee products, with the exception of one sample (*Euphorbia orientalis* honey from Tindouf) with slightly high acidity levels.

In addition, the level of contamination did not appear to be critical because the concentration of contaminants was very low and under the EU regulatory limits available for honey. The only exception was found in cyromazine, whose concentration exceeded the EU limit in most samples from Tiaret and Laghouat. In terms of the number of toxicants detected, the *Euphorbia orientalis* honey from Laghouat was the most contaminated samples while the *Eruca sativa* honey from Tiaret was the least contaminated.

According to the dietary exposure assessment, a small amount of Algerian honey can be safely consumed on a daily basis in both European and Algerian diets. In conclusion, it is hoped that the Algerian authorities will monitor beekeeping activities, find appropriate measures to reduce organic pollution, and harmonise and apply the international regulatory framework concerning the chemical safety of honey, in order to obtain honey of ever higher quality.

## Figures and Tables

**Figure 1 foods-13-01413-f001:**
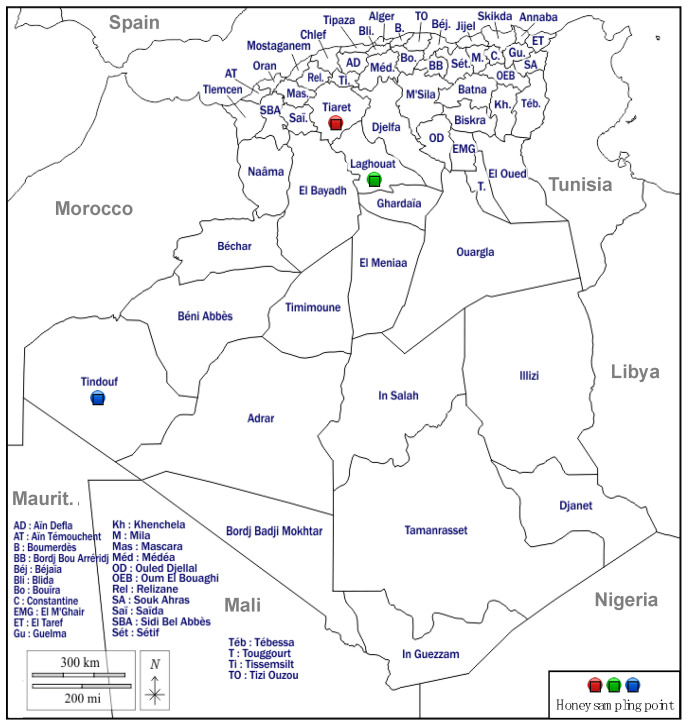
Geographical origin of honey samples considered for the study.

**Figure 2 foods-13-01413-f002:**
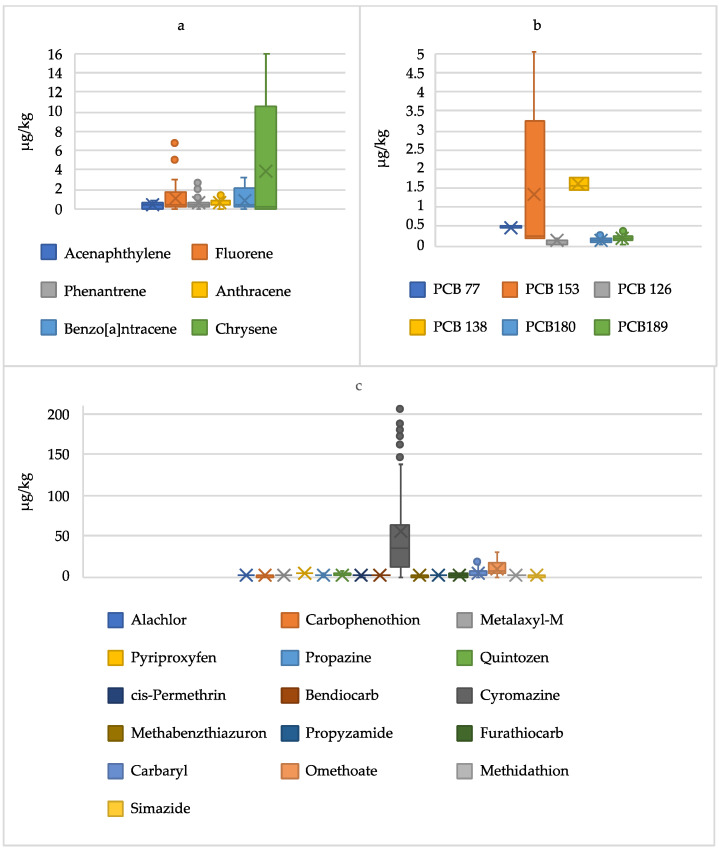
Box-plots of PAH (**a**), PCB (**b**), and pesticide (**c**) concentrations found in Algerian honey.

**Figure 3 foods-13-01413-f003:**
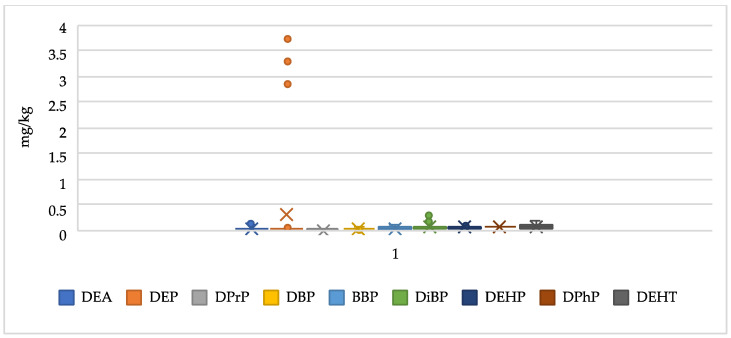
Box-plot illustrating the concentrations of plasticiser residues in Algerian honey.

**Figure 4 foods-13-01413-f004:**
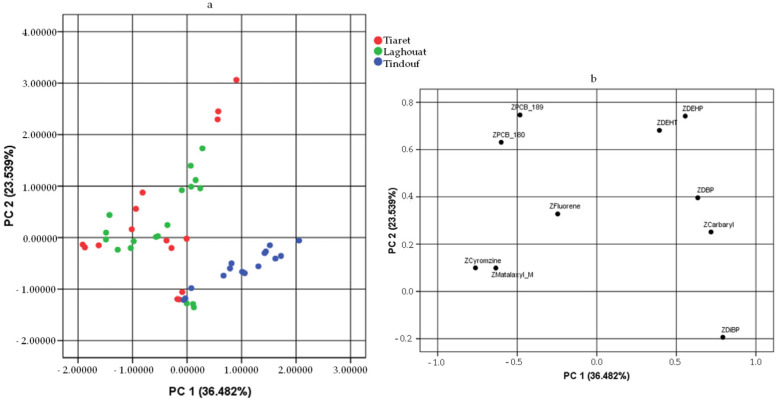
Score plot (**a**) and loading plot (**b**) of PC1 and PC2, explaining honey samples differentiated by geographical origin. For the analysis, only the data on 10 contaminants detected in at least 60% of the samples were considered.

**Figure 5 foods-13-01413-f005:**
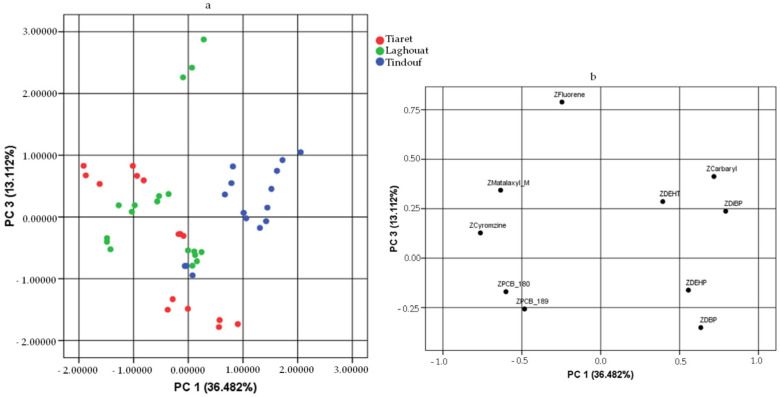
Score plot (**a**) and loading plot (**b**) of PC1 and PC3, explaining honey samples differentiated by geographical origin. For the analysis, only the data on 10 contaminants detected in at least 60% of the samples were considered.

**Table 1 foods-13-01413-t001:** Honey samples and their botanical and geographical origins.

Code	N. of Samples	Botanical Origin	Geographical Origin
M_T_	6	Multifloral	Tiaret
E_T_	3	*Echinops*	Tiaret
ES_T_	3	*Eruca sativa*	Tiaret
ZL_T_	3	*Ziziphus lotus*	Tiaret
BM_T_	3	*Bunium mauritanicum*	Tiaret

TE_L_	3	*Tamarix* and *Euphorbia orientalis*	Laghouat
EO_L_	6	*Euphorbia orientalis*	Laghouat
EG_L_	3	*Eucaliptus globulus*	Laghouat
M_L_	3	Multifloral	Laghouat
Z_L_	6	*Ziziphus lotus*	Laghouat

E_D_	3	*Echinops*	Tindouf
ES_D_	6	*Eruca sativa*	Tindouf
EO_D_	3	*Euphorbia orientalis*	Tindouf
PH_D_	3	*Peganum harmala*	Tindouf
Total	54		

**Table 2 foods-13-01413-t002:** GC-MS instrument operating conditions.

Shimadzu GCMS-TQ8030	Pesticide, PCB, and PAH Analysis	PAE and NPP Analysis
Column	Supelco SLB-5 ms (30 m × 0.25 mm i.d., 0.25 μm film thickness of stationary phase)	
Carrier gas flow rate (He)	0.50 mL/min	0.65 mL/min
Program temperature	60 °C for 1 min, 15 °C/min until 150 °C, 10 °C/min until 270 °C, 2 °C/min until 300 °C	8 °C/min until 190 °C (5 min hold), 8 °C/min until 240 °C (5 min hold), 8 °C/min until 315 °C
Injector temperature	250 °C	
Injection volume	1 µL	1 µL
Injection mode	Splitless with a 1:10 split ratio	Splitless with a 1:15 split ratio
Ion source temperature	230 °C	200 °C
Transferline temperature	290 °C	250 °C
Ionization mode	Electronic ionisation (EI), 70 eV	

**Table 3 foods-13-01413-t003:** Instrument operating conditions for bisphenol analysis.

UHPLC-MS/MS	Shimadzu UHPLC–MS/MS 8040
Column	Phenomenex C18 column (100 mm × 2.1 mm i.d., 1.7 μm particle size)
Mobile phase	Water (A) and acetonitrile (B)
Elution gradient	0 min, 20% B; 2 min, 40% B; 6 min, 90% B; 8 min, 20% B
Flow rate	0.4 mL/min
Injection volume	2 µL
Ionisation mode	ESI negative, 10–40 eV
DL temperature	250 °C
CID gas	230 KPa
Gas nebuliser	Nitrogen
Nitrogen flow	3 L/min
Nitrogen pressure	770 KPa
Collision gas	Argon

**Table 4 foods-13-01413-t004:** Physicochemical parameters (moisture, TSS, conductivity, pH, and acidity) of Algerian honey. Values are expressed as means ± standard deviation of three replicates for each sample with the same botanical and geographical origin.

	Moisture (%)	TTS (°Brix)	Conductivity (µS/cm)	pH	Free Acidity (meq/kg)	Combined Acidity (meq/kg)	Total Acidity (meq/kg)
**M_T_**	14.51 ± 1.59 ^a–d^	81.25 ± 1.30	490.65 ± 24.47 ^a,b^	4.26 ± 0.07 ^a,b^	46.67 ± 0.99 ^a,e,f^	0.86 ± 0.03 ^a^	47.54 ± 0.97 ^a,d^
**E_T_**	14.39 ± 0.13 ^a–d^	84.65 ± 1.32	468.55 ± 7.56 ^a,b^	4.37 ± 0.09 ^a,b^	29.54 ± 0.64 ^b,c^	0.87 ± 0.02 ^a^	30.31 ± 0.64 ^b,e,c^
**ES_T_**	14.15 ± 0.16 ^a–d^	82.36 ± 1.58	326.94 ± 6.77 ^a,c^	4.30 ± 0.02 ^a,b^	41.84 ± 0.97 ^a,d,e^	0.86 ± 0.02 ^a^	42.71 ± 0.96 ^a,c,d^
**ZL_T_**	13.66 ± 0.20 ^a–c^	84.09 ± 1.30	525.79 ± 7.74 ^a,b^	4.65 ± 0.07 ^b,d^	26.37 ± 0.73 ^b,c^	0.88 ± 0.03 ^a^	27.25 ± 0.74 ^b,e^
**BM_T_**	16.55 ± 0.13 ^d^	82.22 ± 1.36	439.15 ± 5.55 ^a–c^	4.40 ± 0.06 ^a,b^	34.12 ± 0.75 ^b,d^	4.18 ± 0.04 ^b^	38.30 ± 0.71 ^a,b^
**TE_L_**	16.15 ± 0.08 ^a,b,d^	81.89 ± 1.43	565.43 ± 8.10 ^b,e^	4.41 ± 0.04 ^a,b^	38.09 ± 0.64 ^a,b,d^	0.84 ± 0.02 ^a^	38.93 ± 0.63 ^a,b^
**EO_L_**	14.12 ± 0.76 ^a–c^	83.65 ± 1.29	544.96 ± 187.42 ^b^	4.50 ± 0.09 ^b,f^	29.80 ± 7.23 ^d^	2.12 ± 0.07 ^c^	31.92 ± 7.16 ^b,g^
**EG_L_**	12.51 ± 0.20 ^c^	84.91 ± 1.61	471.05 ± 6.33 ^a,b^	4.39 ± 0.10 ^a–c^	42.11 ± 0.46 ^a,d,e^	5.23 ± 0.03 ^d^	47.34 ± 0.43 ^a,d,f^
**M_L_**	14.03 ± 0.17 ^b–d^	83.07 ± 1.47	428.21 ± 5.79 ^a–c^	4.50 ± 0.11 ^a,b^	31.27 ± 0.50 ^b,d^	4.89 ± 0.03 ^d^	36.16 ± 0.47 ^b,c,f,g^
**Z_L_**	13.16 ± 0.54 ^c^	84.53 ± 1.24	471.75 ± 44.76 ^a,b^	4.86 ± 0.07 ^b^	20.10 ± 2.26 ^c^	2.85 ± 0.33 ^e^	22.95 ± 2.57 ^e^
**E_D_**	13.93 ± 0.12 ^a,c^	83.19 ± 1.16	330.28 ± 5.03 ^a,c^	3.97 ± 0.05 ^a,c,e^	42.33 ± 0.60 ^d,e^	2.57 ± 0.04 ^e^	44.90 ± 0.60 ^a,d,f^
**ES_D_**	15.34 ± 0.98 ^a,b,d^	82.08 ± 1.91	241.48 ± 7.04 ^c^	3.61 ± 0.43 ^e^	30.73 ± 5.99 ^b^	0.86 ± 0.02 ^a^	31.58 ± 5.98 ^b,g^
**EO_D_**	13.79 ±0.20 ^a,c^	82.57 ± 1.02	337.07 ± 6.78 ^a,c,e^	3.93 ± 0.04 ^a,e^	52.96 ± 0.58 ^e^	0.86 ± 0.01 ^a^	53.82 ± 0.57 ^d^
**PH_D_**	12.32 ± 0.09 ^c^	81.97 ± 1.34	376.50 ± 9.07 ^a–c^	4.08 ± 0.07 ^a,d–f^	37.34 ± 0.52 ^b,d,f^	1.74 ± 0.04 ^f^	39.08 ± 0.55 ^a,g^
** *p* ** **-Value**	**<0.01**	**<0.01**	**<0.01**	**<0.01**	**<0.01**	**<0.01**	**<0.01**

^a–g^ Different superscript letters within the same column denote significantly different values for a specific parameter (*p* < 0.01 by post hoc Tukey’s HSD test); same superscript letters denote not significantly different values for a specific parameter (*p* > 0.01 by post hoc Tukey’s HSD test). Bold *p*-values indicate significantly different results at *p* < 0.01 between different types of honey.

**Table 5 foods-13-01413-t005:** Residues of pesticides, PCBs, and PAHs detected in several honey varieties from Algeria. Data are expressed as means ± standard deviation of three replicates for each sample with the same botanical and geographical origin.

Analyte (µg/kg)	Tiaret					Laghouat					Tindouf				*p*-Value
	M_T_	E_T_	ES_T_	ZL_T_	BM_T_	TE_L_	EO_L_	EG_L_	M_L_	Z_L_	E_D_	ES_D_	EO_D_	PH_D_	
**Bendiocarb**	<LOQ	<LOQ	<LOQ	<LOQ	<LOQ	<LOQ	<LOQ	0.20 ± 0.02	<LOQ	<LOQ	<LOQ	<LOQ	<LOQ	<LOQ	**-**
**Carbaryl**	0.94 ± 0.42 ^a^	7.61 ± 0.61 ^b,e^	<LOQ	1.39 ± 0.17 ^a,d^	1.18 ± 0.13 ^a,d^	0.62 ± 0.06 ^a,d^	1.08 ± 1.14 ^a^	9.46 ± 0.87 ^b^	0.67 ±0.06 ^a,d^	1.49 ± 0.92 ^a,d^	15.81 ± 1.48 ^c^	3.91 ± 4.25 ^a,e^	6.20 ± 0.68 ^b,d,e^	4.51 ± 0.47 ^a,b^	**<0.01**
**Furathiocarb**	<LOQ	<LOQ	<LOQ	2.15 ± 0.27	<LOQ	<LOQ	2.35 ± 2.57	<LOQ	<LOQ	<LOQ	<LOQ	<LOQ	<LOQ	<LOQ	**0.89**
**Metalaxyl-M**	0.42 ± 0.10 ^a,b,e^	0.31 ± 0.04 ^b^	0.32 ± 0.02 ^a,b^	0.63 ± 0.07 ^c,e^	1.10 ± 0.09 ^d,f^	0.78 ± 0.08 ^c^	1.26 ± 0.13 ^d^	0.84 ± 0.05 ^c,f^	0.75 ± 0.08 ^c^	0.46 ± 0.08 ^a,b,e^	0.30 ± 0.03 ^a,b^	0.34 ± 0.03 ^a,b^	0.79 ± 0.08 ^c^	0.27 ± 0.02 ^a,b^	**<0.01**
**Quintozen**	<LOQ	0.35 ± 0.04 ^a^	<LOQ	<LOQ	<LOQ	<LOQ	<LOQ	4.86 ± 0.57 ^b^	0.37 ±0.03 ^a^	<LOQ	<LOQ	<LOQ	<LOQ	<LOQ	**<0.01**
**Methabenzthiazuron**	<LOQ	<LOQ	<LOQ	0.27 ± 0.02	<LOQ	<LOQ	0.35 ± 0.36	0.82 ±0.08	<LOQ	<LOQ	<LOQ	<LOQ	<LOQ	<LOQ	**0.06**
**Propazine**	1.18 ± 1.28	<LOQ	<LOQ	0.46 ± 0.06	1.93 ± 0.13	0.47 ± 0.04	1.28 ± 0.15	<LOQ	0.42 ± 0.04	<LOQ	<LOQ	0.30 ±0.29	<LOQ	<LOQ	**<0.01**
**Propyzamide**	<LOQ	0.12 ± 0.02	<LOQ	<LOQ	<LOQ	<LOQ	<LOQ	<LOQ	<LOQ	<LOQ	<LOQ	<LOQ	<LOQ	<LOQ	**-**
**Simazide**	<LOQ	<LOQ	<LOQ	<LOQ	<LOQ	<LOQ	<LOQ	<LOQ	0.69 ±0.05	<LOQ	<LOQ	0.31 ± 0.21	<LOQ	<LOQ	**0.07**
**Cyromazine**	103.60 ± 94.12 ^a,c,d^	40.28 ± 4.37 ^a,b,d^	16.16 ± 1.54 ^a,b^	50.63 ± 4.48 ^a–c^	163.58 ± 16.20 ^c^	55.90 ± 5.48 ^a,b,c^	123.08 ± 9.66 ^c,d,f^	10.32 ± 1.33 ^a,b^	58.38 ± 4.39 ^a,b,c^	12.77 ± 14.02 ^b,e^	0.30 ± 0.04 ^b,e^	43.21 ± 16.24 ^a,b,e,f^	6.48 ± 0.68 ^a,e^	9.94 ± 0.92 ^a,e^	**<0.01**
**Pyriproxyfen**	<LOQ	<LOQ	<LOQ	3.82 ±0.36	<LOQ	<LOQ	<LOQ	<LOQ	<LOQ	<LOQ	<LOQ	<LOQ	<LOQ	<LOQ	**-**
**Alachlor**	<LOQ	<LOQ	<LOQ	0.15 ±0.03 ^a,c^	0.54 ±0.05 ^a–c^	0.14 ± 0.02 ^a^	0.58 ± 0.25 ^c^	0.36 ± 0.03 ^a–c^	0.76 ± 0.06 ^b^	<LOQ	<LOQ	<LOQ	<LOQ	<LOQ	**<0.01**
**Methidathion**	<LOQ	0.22 ± 0.03	<LOQ	<LOQ	<LOQ	<LOQ	<LOQ	<LOQ	<LOQ	<LOQ	<LOQ	<LOQ	<LOQ	<LOQ	**-**
**Omethoate**	<LOQ	<LOQ	<LOQ	<LOQ	4.82 ± 0.44 ^a^	<LOQ	11.55 ± 12.64 ^a,b^	13.52 ± 1.12 ^a,b^	<LOQ	<LOQ	14.56 ± 1.34 ^a,b^	2.87 ± 3.10 ^a^	4.24 ± 0.38 ^a^	27.54 ± 2.44 ^b^	**<0.01**
**Carbophenothion**	<LOQ	<LOQ	<LOQ	<LOQ	<LOQ	<LOQ	<LOQ	<LOQ	<LOQ	0.43 ± 0.48	0.95 ± 0.09	<LOQ	<LOQ	<LOQ	**0.11**
**cis-Permethrin**	<LOQ	<LOQ	<LOQ	<LOQ	<LOQ	0.48 ±0.03	0.29 ±0.29	<LOQ	<LOQ	<LOQ	<LOQ	<LOQ	0.44 ± 0.04	<LOQ	**0.44**
**Acenaphthylene**	0.25 ± 0.25 ^a^	<LOQ	<LOQ	<LOQ	<LOQ	0.20 ± 0.05 ^a^	0.36 ± 0.36 ^a,b^	<LOQ	<LOQ	<LOQ	<LOQ	0.82 ± 0.08 ^b^	0.22 ± 0.02 ^a,b^	<LOQ	**<0.01**
**Anthracene**	<LOQ	1.23 ± 0.19 ^a,c^	<LOQ	0.38 ± 0.02 ^b,d^	0.36 ± 0.03 ^b,d^	<LOQ	0.23 ± 0.22 ^b^	1.55 ± 0.17 ^c^	0.53 ± 0.07 ^b^	<LOQ	0.91 ± 0.09 ^a,b^	0.28 ± 0.28 ^b^	0.46 ± 0.04 ^b,d^	0.48 ± 0.07 ^b,d^	**<0.01**
**Benzo[a]ntracene**	<LOQ	<LOQ	<LOQ	<LOQ	<LOQ	<LOQ	0.24 ± 0.23	<LOQ	<LOQ	1.60 ± 1.40	<LOQ	<LOQ	<LOQ	<LOQ	**0.04**
**Chrysene**	<LOQ	<LOQ	<LOQ	<LOQ	<LOQ	<LOQ	0.11 ± 0.09	<LOQ	<LOQ	7.39 ± 8.02	<LOQ	<LOQ	<LOQ	<LOQ	**0.04**
**Fluorene**	1.33 ± 1.45 ^a^	<LOQ	<LOQ	0.20 ± 0.01 ^a^	1.44 ± 0.09 ^a^	1.81 ± 0.14 ^a^	0.35 ± 0.37 ^a^	5.73 ± 0.93 ^b^	1.56 ± 0.10 ^a^	<LOQ	0.28 ± 0.04 ^a^	0.17 ±0.16 ^a^	0.30 ± 0.03 ^a^	0.26 ± 0.03 ^a^	**<0.01**
**Phenanthrene**	<LOQ	1.16 ± 0.14 ^a^	<LOQ	0.22 ± 0.02 ^b^	0.25 ± 0.03 ^b^	<LOQ	0.30 ± 0.28 ^b^	2.33 ± 0.39 ^c^	0.40 ± 0.07 ^b^	<LOQ	0.43 ±0.04 ^b^	0.24 ± 0.24 ^b^	0.29 ± 0.03 ^b^	0.19 ± 0.02 ^b^	**<0.01**
**PCB 77**	<LOQ	0.48 ± 0.04	<LOQ	<LOQ	<LOQ	<LOQ	<LOQ	<LOQ	<LOQ	<LOQ	<LOQ	<LOQ	<LOQ	<LOQ	**-**
**PCB 126**	<LOQ	<LOQ	<LOQ	<LOQ	<LOQ	<LOQ	<LOQ	<LOQ	<LOQ	<LOQ	<LOQ	0.11 ± 0.06	0.18 ± 0.02	<LOQ	**0.09**
**PCB 138**	<LOQ	<LOQ	<LOQ	<LOQ	<LOQ	<LOQ	<LOQ	1.59 ± 0.16	<LOQ	<LOQ	<LOQ	<LOQ	<LOQ	<LOQ	**-**
**PCB 153**	<LOQ	<LOQ	<LOQ	<LOQ	<LOQ	<LOQ	0.28 ± 0.05 ^a^	4.67 ± 0.41 ^b^	0.23 ± 0.03 ^a^	<LOQ	<LOQ	<LOQ	<LOQ	<LOQ	**<0.01**
**PCB 180**	0.28 ± 0.05 ^a,b^	0.37 ± 0.03 ^a^	0.17 ± 0.02 ^a,b^	<LOQ	0.25 ± 0.03 ^a,b^	0.14 ± 0.01 ^a,b^	0.36 ± 0.04 ^a^	0.27 ± 0.02 ^a,b^	0.22 ± 0.03 ^a,b^	0.17 ± 0.04 ^b^	<LOQ	0.13 ± 0.13 ^b^	<LOQ	<LOQ	**<0.01**
**PCB 189**	0.29 ± 0.06 ^a,b^	0.43 ± 0.04 ^b^	0.17 ± 0.02 ^a,c^	<LOQ	0.16 ± 0.01 ^a,c^	0.12 ± 0.02 ^c^	0.18 ± 0.02 ^a,c^	0.16 ± 0.02 ^a,c^	0.14 ± 0.01 ^a,c^	0.14 ± 0.11 ^c^	<LOQ	<LOQ	<LOQ	<LOQ	**<0.01**

^a–f^ Different superscript letters within the same line denote significantly different values for a specific parameter (*p* < 0.01 by post hoc Tukey’s HSD test); same superscript letters denote not significantly different values for a specific parameter (*p* > 0.01 by post hoc Tukey’s HSD test). Bold *p*-values indicate significantly different results at *p* < 0.01 between different honey samples.

**Table 6 foods-13-01413-t006:** Residues of plasticisers (PAEs and NPPs) detected in several honey samples from Algeria. Data are expressed as means ± standard deviation of three replicates for each sample with the same botanical and geographical origins.

Analyte (mg/kg)	Tiaret					Laghouat					Tindouf				*p*-Value
	M_T_	E_T_	ES_T_	ZL_T_	BM_T_	TE_L_	EO_L_	EG_L_	M_L_	Z_L_	E_D_	ES_D_	EO_D_	PH_D_	
**DEP**	0.014 ± 0.014	<LOQ	0.038 ± 0.012	0.023 ± 0.006	0.034 ± 0.013	0.026 ± 0.009	0.013 ± 0.013	0.021 ± 0.002	<LOQ	1.656 ± 1.808	<LOQ	<LOQ	<LOQ	<LOQ	**0.02**
**DPrp**	<LOQ	<LOQ	<LOQ	<LOQ	<LOQ	<LOQ	<LOQ	<LOQ	<LOQ	<LOQ	<LOQ	0.016 ± 0.016	<LOQ	<LOQ	**-**
**DBP**	0.048 ± 0.006 ^a^	0.073 ± 0.005 ^b,c^	0.097 ± 0.007 ^b^	0.038 ± 0.007 ^a,c^	0.042 ± 0.006 ^a,c^	0.037 ± 0.006 ^a,c^	0.037 ± 0.005 ^a^	0.041 ± 0.006 ^a,c^	<LOQ	0.041 ± 0.006 ^a^	0.048 ± 0.005 ^a,c^	0.055 ± 0.014 ^b,c^	0.037 ± 0.003 ^a,c^	0.044 ± 0.004 ^a,c^	**<0.01**
**DiBP**	0.036 ± 0.005 ^a^	0.070 ± 0.009 ^a,c^	0.042 ± 0.006 ^a,d^	0.040 ± 0.006 ^a,d^	0.050 ± 0.007 ^a,d^	0.039 ± 0.006 ^a,d^	0.052 ± 0.010 ^a,d^	0.063 ± 0.006 ^c,d^	0.040 ± 0.008 ^a,d^	0.058 ± 0.023 ^a,d^	0.266 ± 0.032 ^b^	0.175 ± 0.141 ^b–d^	0.194 ± 0.021 ^b–d^	0.232 ± 0.027 ^b,c^	**<0.01**
**BBP**	<LOQ	0.041 ± 0.006	<LOQ	<LOQ	<LOQ	<LOQ	<LOQ	<LOQ	0.115 ±0.019	0.020 ± 0.020	<LOQ	<LOQ	<LOQ	<LOQ	**<0.01**
**DPhP**	<LOQ	<LOQ	<LOQ	<LOQ	<LOQ	<LOQ	<LOQ	<LOQ	0.070 ± 0.012	<LOQ	<LOQ	<LOQ	<LOQ	<LOQ	**-**
**DEHP**	0.050 ± 0.009 ^a^	0.118 ± 0.012 ^b^	0.045 ± 0.007 ^a^	0.051 ± 0.004 ^a^	0.053 ± 0.007 ^a^	0.058 ± 0.004 ^a^	0.048 ± 0.008 ^a^	0.073 ± 0.007 ^a^	0.049 ± 0.009 ^a^	0.065 ± 0.024 ^a^	0.070 ± 0.005 ^a^	0.058 ± 0.013 ^a^	0.068 ± 0.004 ^a^	0.073 ± 0.008 ^a^	**<0.01**
**DEA**	0.100 ± 0.108 ^a,b^	0.175 ± 0.025 ^b^	<LOQ	0.027 ± 0.005 ^a,b^	0.047 ± 0.009 ^a,b^	<LOQ	0.068 ± 0.037 ^a,b^	0.033 ± 0.006 ^a,b^	0.046 ± 0.008 ^a,b^	0.013 ± 0.013 ^a^	<LOQ	0.020 ± 0.021 ^a^	<LOQ	<LOQ	**<0.01**
**DEHT**	0.042 ± 0.012 ^a^	0.128 ± 0.019 ^a,b^	0.038 ± 0.009 ^a,b^	0.048 ± 0.009 ^a,b^	0.139 ± 0.015 ^b^	0.103 ± 0.021 ^a,b^	0.053 ± 0.018 ^a,b^	0.144 ± 0.017 ^a,b^	0.055 ± 0.012 ^a,b^	0.102 ± 0.077 ^a,b^	0.108 ± 0.008 ^a,b^	0.076 ± 0.033 ^a,b^	0.094 ± 0.013 ^a,b^	0.089 ± 0.014 ^a,b^	**<0.01**

^a–d^ Different superscript letters within the same line denote significantly different values for a specific parameter (*p* < 0.01 by post hoc Tukey’s HSD test); same superscript letters denote not significantly different values for a specific parameter (*p* > 0.01 by post hoc Tukey’s HSD test). Bold *p*-values indicate significantly different results at *p* < 0.01 between different honeys.

**Table 7 foods-13-01413-t007:** Maximum and minimum values of EDIs (µg/kg_bw_/day or mg/kg_bw_/day) and HQs calculated for Algerian honey consumed daily by typical-sized (70 kg) adult consumers both from Algeria and Europe.

	Algeria				Europe			
	EDI_min_	HQ	EDI_max_	HQ	EDI_min_	HQ	EDI_max_	HQ
	** *Pesticides* **							
**Bendiocarb ***	9.43 × 10^−7^	<1	9.43 × 10^−7^	<1	4.54 × 10^−6^	<1	4.54 × 10^−6^	<1
**Carbaryl ***	2.92 × 10^−6^	<1	7.45 × 10^−5^	<1	1.41 × 10^−5^	<1	3.59 × 10^−4^	<1
**Furathiocarb ***	1.01 × 10^−5^	<1	1.10 × 10^−5^	<1	4.88 × 10^−5^	<1	5.32 × 10^−5^	<1
**Metalaxyl-M ***	1.27 × 10^−6^	<1	5.94 × 10^−6^	<1	6.13 × 10^−6^	<1	2.86 × 10^−5^	<1
**Quintozen ***	1.65 × 10^−6^	<1	2.29 × 10^−5^	<1	7.95 × 10^−6^	<1	1.10 × 10^−4^	<1
**Methabenzthiazuron ***	1.60 × 10^−6^	<1	3.87 × 10^−6^	<1	6.13 × 10^−6^	<1	1.86 × 10^−5^	<1
**Propazine ***	1.32 × 10^−6^	<1	9.10 × 10^−6^	<1	6.36 × 10^−6^	<1	4.38 × 10^−5^	<1
**Propyzamide ***	5.66 × 10^−7^	<1	5.66 × 10^−7^	<1	2.73 × 10^−6^	<1	2.73 × 10^−6^	<1
**Simazide ***	1.37 × 10^−6^	<1	3.25 × 10^−6^	<1	6.59 × 10^−6^	<1	1.57 × 10^−5^	<1
**Cyromazine ***	1.41 × 10^−6^	<1	7.71 × 10^−4^	<1	6.81 × 10^−6^	<1	3.72 × 10^−3^	<1
**Pyriproxyfen ***	1.80 × 10^−5^	<1	1.80 × 10^−5^	<1	8.68 × 10^−5^	<1	8.68 × 10^−5^	<1
**Alachlor ***	6.60 × 10^−7^	<1	3.58 × 10^−6^	<1	3.18 × 10^−6^	<1	1.73 × 10^−5^	<1
**Methidathion ***	1.04 × 10^−6^	<1	1.04 × 10^−6^	<1	5.00 × 10^−6^	<1	5.00 × 10^−6^	<1
**Omethoate ***	1.34 × 10^−5^	<1	1.30 × 10^−4^	<1	6.45 × 10^−5^	<1	6.26 × 10^−4^	<1
**Carbophenothion ***	2.03 × 10^−6^	<1	4.48 × 10^−6^	<1	9.77 × 10^−6^	<1	2.16 × 10^−5^	<1
**cis-Permethrin ***	1.27 × 10^−6^	<1	2.26 × 10^−6^	<1	6.13 × 10^−6^	<1	1.09 × 10^−5^	<1
	** *Plasticisers* **							
**DEA ****	5.66 × 10^−8^	-	8.25 × 10^−7^	-	2.73 × 10^−7^	-	3.98 × 10^−6^	-
**DEP ****	5.66 × 10^−8^	<1	7.81 × 10^−3^	<1	2.73 × 10^−7^	<1	3.76 × 10^−2^	<1
**DPrp ****	7.07 × 10^−8^	-	7.07 × 10^−8^	-	3.41 × 10^−7^	-	3.41 × 10^−7^	-
**DiBP ****	1.70 × 10^−7^	-	1.25 × 10^−6^	-	8.18 × 10^−7^	-	6.04 × 10^−6^	-
**DBP ****	1.74 × 10^−7^	<1	4.57 × 10^−7^	<1	8.40 × 10^−7^	<1	2.20 × 10^−6^	<1
**BBP ****	8.96 × 10^−8^	-	5.42 × 10^−7^	-	4.32 × 10^−7^	-	2.61 × 10^−6^	-
**DEHP ****	2.12 × 10^−7^	<1	5.56 × 10^−7^	<1	1.02 × 10^−6^	<1	2.68 × 10^−6^	<1
**DPhP ****	3.30 × 10^−7^	-	3.30 × 10^−7^	-	1.59 × 10^−6^	-	1.59 × 10^−6^	-
**DEHT ****	1.79 × 10^−7^	-	6.79 × 10^−7^	-	8.63 × 10^−7^	-	3.27 × 10^−6^	-

* µg/kg_bw_/day. ** mg/ kg_bw_/day.

## Data Availability

The original contributions presented in the study are included in the article and Appendix A, further inquiries can be directed to the corresponding authors.

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
