# Peer review of "Study of Physicochemical Quality and Organic Contamination in Algerian Honey"

_foods, 2024, doi:10.3390/foods13091413_

Round 1
Reviewer 1 Report
Comments and Suggestions for Authors
The manuscript is interesting and gain some new and significant knowledge in the field of quality and safety of honey. The literature review is done properly. Sampling and all applied methodology are adequate. Results are well written; however, I suggest presenting most important results with some kind of figures, having in mind it is very hard to recognize most significant results in lot of data presented in Tables. Consider making boxplot figures or heatmaps.
My other concern is the local (Algeria) tone of discussion and conclusions. The study was conducted on Algerian honey but, having in mind this is international journal, I am suggesting to authors to try to discuss and make conclusions from their findings in general and worldwide important conclusions.
Author Response
The manuscript is interesting and gain some new and significant knowledge in the field of quality and safety of honey. The literature review is done properly. Sampling and all applied methodology are adequate. Results are well written; however, I suggest presenting most important results with some kind of figures, having in mind it is very hard to recognize most significant results in lot of data presented in Tables. Consider making boxplot figures or heatmaps.
R: Dear Reviewer 1 thank you for your valuable comments! We did our best to address your suggestions and improve the overall quality of the manuscript. The suggestion has been accepted and boxplot figures were used to better graphically present the results.
My other concern is the local (Algeria) tone of discussion and conclusions. The study was conducted on Algerian honey but, having in mind this is international journal, I am suggesting to authors to try to discuss and make conclusions from their findings in general and worldwide important conclusions.
R: In the discussion section, data have been already discussed in reference to previous and recent inherent literature coming from not only African areas but also major European producer countries. Additionally, the suggestion has been accepted by highlighting the global relevance of the study in the conclusion section.

Reviewer 2 Report
Comments and Suggestions for Authors
In this study, physicochemical quality and organic contamination in Algerian honey were monitored and analyzed. The study is meaningful and may provide the basic information for the quality control of honey in Algerian. Some minor comments are as follows:
1. The content of this paper has too high duplication rate with the previous articles published by this research team, so it is suggested to modify it to reduce the duplication rate
2. It is recommended to use principal component analysis (PCA) should to identify the differences of organic contaminants in honey from different regions of Algerian and to systematically analyze the characteristic these contaminants.
Author Response
In this study, physicochemical quality and organic contamination in Algerian honey were monitored and analyzed. The study is meaningful and may provide the basic information for the quality control of honey in Algerian. Some minor comments are as follows:
- The content of this paper has too high duplication rate with the previous articles published by this research team, so it is suggested to modify it to reduce the duplication rate
R: Dear Reviewer 2 thank you for your valuable comments! We did our best to address your suggestions and improve the overall quality of the manuscript. The suggestion has been accepted and the article was modified to obtain a lower duplication rate.
- It is recommended to use principal component analysis (PCA) should to identify the differences of organic contaminants in honey from different regions of Algerian and to systematically analyze the characteristic these contaminants.
R: The suggestion has been accepted and a PCA was performed by considering only those contaminants present in at least 60% of samples.

Reviewer 3 Report
Comments and Suggestions for Authors
Nice work. Here are my suggestions for authors.
Line 16: I would not use term "substance" for honey since, chemically, it is completely incorrect. Suggest to replace with "mixture" or "natural product". Please replace at all places in text.
Line 36: It should be "Apis mellifera L." since there are several species of Apis spp.
Line 50: Please revise this "... of apiculutre... such as". It is hard to understand.
Line 52: "EU regulation" here? Specify.
Lines 53 and 69: Suggest to rewrite as follow: ".... in foods, including honey".
Line 77: "EU" not "UE". Correct.
Line 80: Please specify here value for TDI mentioned.
Line 120: Put all Latin names for plants, provided in the Table 1, in Italic.
Lines 165-166: Please use molarity as SI unit for concentration although acidity is usually calculated with normality. In case of HCl N will be the same with M.
Lines 294-301: typos- "kg" not "Kg" for mass unit. Correct all.
Lines 324-354: Put all Latina names in Italic here. Also, one name (Line 330) is incorrectly written. It should be E. orientalis?
Lines 330-354: Same typos as in the Lines 204-301.
Line 385: "... the north-west..." Correct.
Lines 411-415: typos with "Kg" and Latin names as previous.
Line 418: I would say "honeys" in plural here. Correct.
Line 439: typo "kg". Correct.
Line 462: "EU" not "UE". Correct.
Kind regards.
Comments on the Quality of English Language
English is fine. Some minor corrections are requested in Section for authors.
Author Response
Reviewer 3°
Nice work. Here are my suggestions for authors.
Dear Reviewer 3 thank you for your valuable comments! We did our best to address your suggestions and improve the overall quality of the manuscript.
Line 16: I would not use term "substance" for honey since, chemically, it is completely incorrect. Suggest to replace with "mixture" or "natural product". Please replace at all places in text.
R: The suggestion has been accepted.
Line 36: It should be "Apis mellifera L." since there are several species of Apis spp.
R: The suggestion has been accepted.
Line 50: Please revise this "... of apiculutre... such as". It is hard to understand.
R: The suggestion has been accepted.
Line 52: "EU regulation" here? Specify.
R: The suggestion has been accepted. This regulation is published in the Official Journal of the European Union as Regulation of European Community (EC), forerunner of European Union.
Lines 53 and 69: Suggest to rewrite as follow: ".... in foods, including honey".
R: The suggestion has been accepted.
Line 77: "EU" not "UE". Correct.
R: The suggestion has been accepted.
Line 80: Please specify here value for TDI mentioned.
R: The suggestion has been accepted.
Line 120: Put all Latin names for plants, provided in the Table 1, in Italic.
R: The suggestion has been accepted.
Lines 165-166: Please use molarity as SI unit for concentration although acidity is usually calculated with normality. In case of HCl N will be the same with M.
R: The suggestion has been accepted.
Lines 294-301: typos- "kg" not "Kg" for mass unit. Correct all.
R: The suggestion has been accepted.
Lines 324-354: Put all Latina names in Italic here. Also, one name (Line 330) is incorrectly written. It should be E. orientalis?
R: The suggestion has been accepted.
Lines 330-354: Same typos as in the Lines 204-301.
R: The suggestion has been accepted.
Line 385: "... the north-west..." Correct.
R: The suggestion has been accepted.
Lines 411-415: typos with "Kg" and Latin names as previous.
Line 418: I would say "honeys" in plural here. Correct.
R: The suggestion has been accepted.
Line 439: typo "kg". Correct.
R: The suggestion has been accepted.
Line 462: "EU" not "UE". Correct.
R: The suggestion has been accepted.
